# Transglutaminase 2 Facilitates Murine Wound Healing in a Strain-Dependent Manner

**DOI:** 10.3390/ijms241411475

**Published:** 2023-07-14

**Authors:** Ting W. Yiu, Sara R. Holman, Xenia Kaidonis, Robert M. Graham, Siiri E. Iismaa

**Affiliations:** 1Victor Chang Cardiac Research Institute, Darlinghurst, NSW 2010, Australia; tingwaiyiu@gmail.com (T.W.Y.); holmansarar@gmail.com (S.R.H.); x.kaidonis@victorchang.edu.au (X.K.); 2School of Clinical Medicine, UNSW Medicine and Health, University of New South Wales Sydney, Kensington, NSW 2052, Australia

**Keywords:** TG2, wound healing, scratch wound, cytoskeletal dynamics, mouse strain, integrin, syndecan, murine embryonic fibroblasts, bupivacaine, β-sandwich-core domain

## Abstract

Transglutaminase 2 (TG2) plays a role in cellular processes that are relevant to wound healing, but to date no studies of wound healing in TG2 knockout mice have been reported. Here, using 129T2/SvEmsJ (129)- or C57BL/6 (B6)-backcrossed TG2 knockout mice, we show that TG2 facilitates murine wound healing in a strain-dependent manner. Early healing of in vivo cutaneous wounds and closure of in vitro scratch wounds in murine embryonic fibroblast (MEF) monolayers were delayed in 129, but not B6, TG2 knockouts, relative to their wild-type counterparts, with wound closure in 129 being faster than in B6 wild-types. A single dose of exogenous recombinant wild-type TG2 to 129 TG2^−/−^ mice or MEFs immediately post-wounding accelerated wound closure. Neutrophil and monocyte recruitment to 129 cutaneous wounds was not affected by *Tgm2* deletion up to 5 days post-wounding. *Tgm2* mRNA and TG2 protein abundance were higher in 129 than in B6 wild-types and increased in abundance following cutaneous and scratch wounding. *Tgm1* and factor XIIA (*F13A*) mRNA abundance increased post-wounding, but there was no compensation by TG family members in TG2^−/−^ relative to TG2^+/+^ mice in either strain before or after wounding. 129 TG2^+/+^ MEF adhesion was greater and spreading was faster than that of B6 TG2^+/+^ MEFs, and was dependent on syndecan binding in the presence, but not absence, of RGD inhibition of integrin binding. Adhesion and spreading of 129, but not B6, TG2^−/−^ MEFs was impaired relative to their wild-type counterparts and was accelerated by exogenous addition or transfection of TG2 protein or cDNA, respectively, and was independent of the transamidase or GTP-binding activity of TG2. Rho-family GTPase activation, central to cytoskeletal organization, was altered in 129 TG2^−/−^ MEFs, with delayed RhoA and earlier Rac1 activation than in TG2^+/+^ MEFs. These findings indicate that the rate of wound healing is different between 129 and B6 mouse strains, correlating with TG2 abundance, and although not essential for wound healing, TG2 facilitates integrin- and syndecan-mediated RhoA- and Rac1-activation in fibroblasts to promote efficient wound contraction.

## 1. Introduction

Rapid and efficient tissue repair after wounding is a primary biological response critical to survival of the organism. A complex and dynamic process, it involves the highly coordinated interaction of various cells with their surrounding extracellular matrix (ECM). These responses are orchestrated by a variety of cytokines and growth factors in the ordered and overlapping phases of inflammation, cell proliferation and tissue remodelling [1]. Soon after wounding, a provisional ECM ‘plug’ is formed that provides a scaffold for cell migration. This is followed by invasion of inflammatory cells, and then fibroblasts and capillaries, to form contractile granulation tissue. Concomitantly, re-epithelialization commences and the epidermal edges migrate to cover the wounded surface [1]. Fibroblasts, dispersed throughout the connective tissue, play a vital role in wound healing contributing importantly to the de novo elaboration of ECM [2]. Fibroblast adhesion, spreading and migration, all integral to wound contraction, involve cooperative signaling between integrin and syndecan cell surface receptors to regulate protein kinase and Rho-family GTPase signaling [3]. The complex interaction of these pathways enables the cellular responses that are critical for efficient wound healing (see [4,5], for example).

Transglutaminase 2 (TG2) is a multi-functional protein [6,7]. As a member of a family of transamidating enzymes, TG2 catalyzes post-translational protein-modification (or cross-linking) as a result of isopeptide bond formation between an intrachain glutamine and lysine residue, or by glutamine-derivatization with a primary amine. Substrates for the transamidase reaction include ECM proteins: collagen, fibrin, fibronectin, fibrinogen, laminin/nidogen complexes, osteonectin, osteopontin and vitronectin [8,9,10,11,12,13,14]. TG2 functions intracellularly to bind and hydrolyze GTP [15], an activity that is mutually exclusive to its transamidase activity [16,17]. TG2 can also act as an adapter molecule that facilitates interaction between fibronectin, β1 or β3-integrins and syndecan-4 [18,19,20,21]. TG2 is distributed widely as a result of constitutive expression by fibroblasts, smooth muscle cells and endothelial cells, as well as being expressed by a number of organ-specific cell types [6]. At the subcellular level, TG2 is found in the cytosol, in association with plasma and nuclear membranes, and extracellularly, where it is attached to the ECM as a matricellular protein [22,23].

The role of TG2 in cellular processes that are relevant to wound healing, such as cell adhesion, spreading and migration, has been widely studied and well-reviewed [24,25]. TG2 expression and activity are increased at sites of neovascularization and invasion of the fibrin matrix and, later, in the granulation tissue matrix where it may crosslink ECM substrates [26,27,28]. Studies of cultured cells have provided evidence for a role of TG2 in adhesion, spreading and motility [18,23,29,30,31,32]. However, no studies on the effect of TG2 on mammalian wound healing have been reported to date. Here, using TG2 knockout mice on a 129T2/SvEmsJ (129) or C57BL/6 (B6) strain background [33], we show that the rate of wound healing differs between 129 and B6 mouse strains coincident with TG2 abundance, and although not essential for wound healing, TG2 facilitates integrin- and syndecan-mediated RhoA- and Rac1-activation in fibroblasts to promote efficient murine wound contraction with no compensation by other TG family members.

## 2. Results

### 2.1. Early Healing of Cutaneous Wounds Was Slower in 129 TG2^−/−^ Mice Than in 129 TG2^+/+^, B6 TG2^+/+^ or B6 TG2^−/−^ Mice

Wound healing in mice subjected to a circular skin punch biopsy wound was slower in 129 TG2^−/−^ relative to 129 TG2^+/+^ mice for the first 5 days after wound healing (Figure 1A,B), corresponding to the inflammatory and early-to-mid proliferative phases of wound healing [34]. On the first day post-wounding, the wound area in 129 TG2^−/−^ mice was reduced to ~70% of the day 0 area compared to ~50% of the day 0 area in 129 TG2^+/+^ mice (*p* < 0.001). By day 5, the wounds in 129 TG2^−/−^ mice had contracted to ~50% of the day 0 area compared to ~30% in TG2^+/+^ mice (*p* < 0.001). From day 6 onwards, there was no significant difference between the two genotypes in terms of wound size as a fraction of day zero. Complete wound closure in both genotypes was achieved in 11 days, with the integrated wound closure time (area under the wound closure curve) being greater for 129 TG2^−/−^ mice (4.1 ± 0.3, arbitrary units) than for TG2^+/+^ mice (1.9 ± 0.1, *p* < 0.001) (Figure 1C). The initial delay in wound healing in 129 TG2^−/−^ mice was reflected in the integrated wound closure time over the first 5 days of wound healing, being significantly greater in TG2^−/−^ than TG2^+/+^ mice (3.02 ± 0.09 vs. 2.28 ± 0.16, respectively; *p* < 0.001, two-tailed *t* test), but was no different from day 6 onwards (0.39 ± 0.09 in TG2^−/−^ vs. 0.37 ± 0.09 in TG2^+/+^ mice; *p* = ns, two-tailed *t* test).

In contrast, in vivo cutaneous wound healing was not significantly different between B6 TG2^+/+^ and B6 TG2^−/−^ mice (Figure 1D,E). Wound closure in the B6 lines was complete after 10 or 11 days, similar to that for the 129 TG2 lines. Integrated wound closure times were 2.78 ± 0.26 for B6 TG2^−/−^ and 2.96 ± 0.22 for B6 TG2^+/+^ mice (*p* = ns, Figure 1F), which were intermediate between those of 129 TG2^−/−^ (4.11 ± 0.33) and 129 TG2^+/+^ (1.86 ± 0.13; Figure 1C). These results indicate a strain dependent effect of TG2 on wound healing.

### 2.2. Bupivacaine Injection Prior to Skin Punch Biopsy Abrogated the Difference in Wound Healing between 129 TG2^−/−^ and 129 TG2^+/+^ Mice and Delayed Wound Healing

A change to the wound healing assay protocol, involving subcutaneous injection of the long-acting analgesic bupivacaine instead of saline at the site prior to wounding, resulted in a marked increase in wound area size during the inflammatory and early-to-mid proliferative phases (days 1–5) in both 129 TG2^−/−^ and 129 TG2^+/+.^ mice (Figure 2A), and abrogation of the retardation in 129 TG2^−/−^ wound closure relative to 129 TG2^+/+^ observed previously (Figure 1A–C). Complete wound closure in both genotypes was achieved in 11 days, with the integrated wound closure time increasing to 6.4 ± 0.5 in 129 TG2^−/−^ and 6.7 ± 0.32 in 129 TG2^+/+^ mice (*p* = ns, Figure 2B). Given that bupivacaine inhibits cytokine production by activated macrophages [35], these data support involvement of the immune response in efficient wound repair. Subsequent experiments were performed without the use of bupivacaine.

### 2.3. Neutrophil and Monocyte Recruitment to Cutaneous Wounds Was No Different between 129 TG2^−/−^ Mice and TG2^+/+^ Mice up to 5 Days Post-Wounding

Neutrophils and monocytes are the predominant cell types during the inflammatory and cell proliferation phases of wound healing [36]. Quantitation of neutrophils and monocytes in wound sections showed no significant differences in abundance between 129 TG2^−/−^ and TG2^+/+^ wounds (Figure 3). Neutrophil counts were high on day 1 post-wounding and progressively declined over the next 4 days; monocyte counts were about 4-fold less than neutrophil counts on day 1 post-wounding and slowly doubled over a period of 48 h to become the predominant cell type in the wound, remaining stable for the following 48 h. Thus, neutrophil and monocyte infiltration into the wound is not affected in 129 TG2^−/−^ relative to 129 TG2^+/+^ mice.

### 2.4. mRNA Abundance of Wound-Related Genes Was Increased to the Same Levels in 129 TG2^−/−^ and TG2^+/+^ Mice on Day 2 Post-Wounding

Microarray analysis of total RNA from 129 TG2^−/−^ and TG2^+/+^ wounds collected at the inflammation/early cell proliferation phase on day 2 post-wounding showed that *Tgm2* was the only gene that was significantly different in abundance (~200-fold) between TG2^−/−^ and TG2^+/+^, despite all criteria for validation and controls having been met (see Appendix A for a list of all genes that had changes of at least 0.1 log-fold, *p* < 0.05). To determine if the wound samples harvested contained too much healthy tissue, which might have diluted the ’wounded’ transcriptome, we evaluated the day 2 post-wounding cutaneous samples that were used for microarray analysis as well as unwounded skin samples by quantitative RT-PCR for candidate genes known to be involved in inflammation and wound healing. The innate immune response post-wounding is regulated by toll-like receptors (TLRs), which signal through Myeloid Differentiation Primary Response 88 (MyD88)-dependent and -independent pathways [36]. The MyD88-dependent pathway leads to nuclear factor κB (NF-κB)-dependent transcription of pro-inflammatory cytokines, such as interleukin 6 (IL6) and tumor-necrosis factor alpha (TNF-α); activation of the MyD88-independent pathway is mediated by TICAM1 (Toll/interleukin-1 receptor (TIR) Domain-Containing Adapter Molecule 1), resulting in NF-κB-dependent induction of apoptotic cytokines, such as interferon gamma (IFN-γ) [37]. TG2 induces nuclear translocation of NF-κB by transamidating Inhibitor of NF-κB (I-κB) [38]. Growth factors, such as epidermal growth factor (EGF), fibroblast growth factor (FGF) and transforming growth factor beta (TGF-β), are released by activated macrophages to promote re-epithelialization [1]. Quantitation of the abundance of mRNAs for these genes revealed an increase on day 2 post-wounding, relative to unwounded skin (Figure 4A,B), with no significant difference between 129 TG2^−/−^ and TG2^+/+^ mice (*Ticam1*, ~1.5 fold; *MyD88*, ~2 fold; *Nfkb1*, ~1.4 fold; *Nfkbib*, ~1.6 fold, *Il6*, ~6 fold; *Ifng*, ~5.1 fold; *Tnf*, ~6.5 fold; *Egf*, ~9.5 fold; *Fgf2*, ~13.2 fold; *Tgfb2*, ~2 fold). The abundance of mRNAs for ECM proteins, which promote and regulate wound healing [1], and the proteoglycan cell surface adhesion and migration receptor, syndecan 4, was also increased on day 2 post-wounding, relative to unwounded skin (Figure 4C), with no significant difference between genotypes (*Fndc4*, ~2 fold; *Fndc3a*, ~4 fold; *Col1a1*, ~1.6 fold; *Col1a2*, ~3 fold; *Col3a1*, ~1.25 fold; *Sdc4*, ~3 fold). These data provide evidence that the post-wounding samples reflect a ’wounded’ transcriptome in both genotypes, compared to their respective unwounded samples, and that the ’wounded’ transcriptome response of 129 TG2^−/−^ is not defective relative to that of 129 TG2^+/+^ mice.

In addition, relative to unwounded skin, quantitation of abundance of mRNAs for other TG family members showed that *Tgm1* and factor XIIA (*F13a*) were increased ~1.5 fold on day 2 post-wounding in both genotypes while *Tgm2* mRNA abundance was increased only in 129 TG2^+/+^ mice (~2 fold) (Figure 4D). This confirmed that TG2 mRNA is absent from 129 TG2^−/−^ mice as expected and that there is no compensation by other TGs for the lack of TG2 in 129 TG2^−/−^ mice. Together, these data provide compelling evidence that the delay in early wound healing in 129 TG2^−/−^ relative to 129 TG2^+/+^ mice is not due to a defective ’wounded’ transcriptome but rather, to a direct effect of TG2 protein and/or activity.

### 2.5. A Single Dose of TG2 Protein to Cutaneous Wounds in 129 TG2^−/−^ Mice Restored Rate of Wound Healing to That in 129 TG2^+/+^ Mice

The effect on wound healing of exogenous application of purified TG2 protein was then investigated. Relative to vehicle, a single application on day 0 of TG2 protein to cutaneous wounds of 129 TG2^−/−^ mice significantly increased the rate of wound healing (Figure 5A,C). On day 1 post-wounding, 129 TG2^−/−^ wounds treated with TG2 were reduced to 67% of the day 0 area compared with 76% of the day 0 area of vehicle-treated 129 TG2^−/−^ wounds (*p* < 0.05). A single application on day 0 of TG2 protein to cutaneous wounds of 129 TG2^+/+^ mice had no significant effect, relative to vehicle, on wound healing (Figure 5B,C). There was no significant difference between integrated wound closure times of TG2-treated 129 TG2^−/−^ and vehicle-treated 129 TG2^+/+^ wounds (Figure 5C). This confirms a direct effect of TG2 protein and/or activity on cutaneous wound healing.

### 2.6. The Number of 129 TG2^−/−^ MEFs Adherent to an Fn Matrix Was Less Compared to 129 TG2^+/+^ MEFs, but Was No Different from B6 TG2^+/+^ or B6 TG2^−/−^ MEFs

Fibroblast adhesion, spreading and migration play a vital role in wound contraction and there is evidence that TG2 facilitates adhesion and motility in fibroblasts on a fibronectin (Fn) matrix [21,25]. To dissect this further, TG2^+/+^ and TG2^−/−^ murine embryonic fibroblasts (MEFs) were isolated with the aim of studying their cellular dynamics in vitro. Proliferation rates of TG2^+/+^ and TG2^−/−^ MEFs were similar for 129 and B6 lines (Appendix A), although the cell density at confluence tended to be slightly lower for 129 TG2^−/−^ than for 129 TG2^+/+^ MEFs (Appendix A). Following optimization of conditions for MEF adhesion on Fn (Appendix A), quantitation of adherent cells showed significantly more adherent 129 TG2^+/+^ than 129 TG2^−/−^ MEFs, with no significant difference between the number of adherent 129 TG2^−/−^, B6 TG2^+/+^ or B6 TG2^−/−^ MEFs (Figure 6). At 30 min, when adhesion was starting to plateau, 55% of the total number of 129 TG2^+/+^ cells seeded were adherent compared to ~35% of 129 TG2^−/−^, B6 TG2^+/+^ or B6 TG2^−/−^ MEFs. These results indicate, firstly, that adhesion of 129 TG2^−/−^ MEFs is defective relative to 129 TG2^+/+^ MEFs and secondly, that 129 TG2^+/+^ MEFs have a greater capacity to adhere than B6 TG2^+/+^ or B6 TG2^−/−^ MEFs.

### 2.7. The Number of MEFs Adhered to Fn Was Increased by the Exogenous Addition of Recombinant TG2 and Required the β-Sandwich and Core Domains but Not the Transamidase or GTP-Binding Activity of TG2

The addition of recombinant wild-type TG2 to Fn-coated plates resulted in a dose-dependent increase in the number of adherent MEFs, with significantly more adherent 129 TG2^+/+^ than 129 TG2^−/−^ MEFs, but no significant difference between the number of adherent 129 TG2^−/−^, B6 TG2^+/+^ or B6 TG2^−/−^ MEFs (Figure 7A). The addition of 20 μg/cm^2^ wild-type TG2 rescued the number of adherent 129 TG2^−/−^ MEFs to that of 129 TG2^+/+^ MEFs plated on Fn with no exogenous wild-type TG2 added. The number of adherent 129 TG2^+/+^ MEFs plateaued at 20 μg/cm^2^ wild-type TG2 with adhesion of almost 90% of the total number of 129 TG2^+/+^ cells seeded. This same level of adhesion of 129 TG2^−/−^ MEFs was achieved with 60 μg/cm^2^ wild-type TG2. The adhesion profiles of 129 TG2^+/+^ and 129 TG2^−/−^ MEFs in response to the addition of recombinant W241A (transamidase-deficient [39]) or R579A (GTP-binding-deficient [40]) TG2 to Fn-coated plates (Figure 7B,C) were similar to those observed with wild-type TG2 (Figure 7A). A truncated mutant form of TG2 that consists only of the β-sandwich-core domain [41], but not TG2 β-sandwich or core [41] domains alone, was able to substitute for full-length TG2 to increase the number of adherent 129 TG2^+/+^ and 129 TG2^−/−^ MEFs (Appendix A). This demonstrates that the number of adherent MEFs is increased by the exogenous addition of TG2 to the Fn matrix and that adhesion requires the N-terminal β-sandwich and core domains of TG2 but does not require either the transamidase or the GTP-binding activity of TG2.

### 2.8. Adhesion of 129 TG2^+/+^ or 129 TG2^−/−^ MEFs Was Dependent on Syndecan Binding in the Presence, but Not Absence, of RGD Peptides, and the Exogenous Addition of Recombinant Wild-Type TG2 to an Fn Matrix Rescued RGD-Impaired MEF Adhesion

Fibroblast adhesion on Fn is mediated by members of the integrin receptor family (α4β1, α5β1, β3) and the syndecan co-receptor family (Synd4) [3], and TG2 is known to interact with Fn [42], integrins [18] and syndecan-4 [20]. The relative contribution of TG2, integrins and syndecans to 129 MEF adhesion on Fn was assessed with the Fn synthetic peptide GRGDTP (RGD), a competitive inhibitor of the Fn-integrin interaction, in the absence or presence of heparin, which inhibits TG2 interaction with the heparan sulphate chains of syndecan receptors. RGD addition reduced the number of adherent 129 TG2^+/+^ MEFs by ~25% and the number of adherent 129 TG2^−/−^ MEFs by ~50% (Figure 8A). The exogenous addition of recombinant wild-type TG2 (20, 40 60 or 80 μg/cm^2^) to the Fn matrix dose-dependently overcame the RGD-induced reduction in adherent 129 MEFs until a plateau was reached at 40 μg/cm^2^ or 60 μg/cm^2^ TG2 for 129 TG2^+/+^ or TG2^−/−^ MEFs, respectively, where the numbers of adherent cells were equivalent to those with no RGD peptide. The addition of the non-specific GRADSP (RAD) negative control had no significant effect on the adhesion capacity of 129 TG2^+/+^ or TG2^−/−^ MEFs on Fn ± wild-type TG2 (Figure 8A). These data are consistent with an Fn-TG2 matrix rescuing RGD-impaired cell adhesion [20,21,43].

The addition of heparin, alone, had no significant effect on the adhesion capacity of 129 TG2^+/+^ or TG2^−/−^ MEFs on Fn ± TG2 relative to no heparin treatment (Figure 8B). However, the addition of both RGD peptides and heparin reduced the number of adherent 129 TG2^+/+^ MEFs by ~50% and the number of adherent 129 TG2^−/−^ MEFs by ~75% (Figure 8B). The addition of wild-type TG2 to the Fn matrix was unable to increase MEF adhesion capacity in the presence of RGD and heparin. These data are consistent with MEF adhesion on an Fn-TG2 matrix being dependent on syndecan binding in the presence, but not the absence, of RGD peptides.

### 2.9. Spreading of 129 TG2^−/−^, B6 TG2^+/+^ and B6 TG2^−/−^ MEFs Was Equivalent but Delayed Relative to 129 TG2^+/+^ MEFs, and Was Accelerated in All Genotypes by Either the Exogenous Addition of Recombinant TG2 or Transfection of cDNAs Encoding TG2, Independent of Transamidase or GTP-Binding Activity

Once fibroblasts adhere by attachment onto the substratum, they begin to spread by extending cytoplasmic processes [3]. Quantitation of cell spreading showed a significant delay in cell spreading of 129 TG2^−/−^ MEFs relative to 129 TG2^+/+^ MEFs, with no significant difference in the rate of cell spreading between 129 TG2^−/−^, B6 TG2^+/+^ or B6 TG2^−/−^ MEFs (Figure 9). At 30 min, 129 TG2^−/−^ and 129 TG2^+/+^ MEFs were rounded, actin staining was diffuse with no visible actin fibers and the mean 129 TG2^−/−^ MEF cell area was half that of 129 TG2^+/+^ MEFs (Figure 9A,B). Cells areas increased progressively over time. At 90 min, thick actin stress fibers were observed across the cell body in both genotypes and the mean 129 TG2^−/−^ MEF cell area was 80% of that of 129 TG2^+/+^ MEFs. Exogenous addition of recombinant wild-type TG2 to the Fn matrix increased the rate of 129 TG2^−/−^ MEF spreading to that of 129 TG2^+/+^ MEFs with Fn alone in the matrix, and also accelerated 129 TG2^+/+^ MEF spreading relative to 129 TG2^+/+^ MEFs without added TG2 in the matrix. The exogenous addition of TG2 to the matrix also increased the rate of B6 MEF spreading, but only to that of 129 TG2^+/+^ MEFs with Fn alone in the matrix (Figure 9C,D). Transamidase-deficient or GTP-binding-deficient TG2 mutants were as effective as wild-type TG2 in accelerating 129 MEF spreading when added exogenously to the Fn matrix (Figure 9E) or following transfection of encoding cDNAs (Figure 9F). Mock-transfected cells spread to the same extent as untreated 129 TG2^−/−^ MEFs, indicating that neither the plasmid nor the transfection procedure affected cell spreading. These results demonstrate that spreading of 129 TG2^−/−^, B6 TG2^+/+^ and B6 TG2^−/−^ MEFs is equivalent but delayed relative to 129 TG2^+/+^ MEFs and that the exogenous addition of recombinant TG2 protein or transfection of cDNAs encoding TG2 accelerated spreading, independent of transamidase or GTP-binding activity.

### 2.10. Activation Dynamics of the GTPases RhoA and Rac1 That Control Cell Adhesion and Spreading on Fn Were Altered in 129 TG2^−/−^ Relative to TG2^+/+^ MEFs

Fibroblast adhesion on Fn-coated plates activates the small GTPases, RhoA and Rac1 [44,45,46]. Quantitation of the relative levels of activation of these GTPases revealed altered dynamics in 129 TG2^−/−^ relative to TG2^+/+^ MEFs (Figure 10). Although levels of GTP-bound RhoA increased significantly in TG2^−/−^ MEFs in response to adhesion over time, this increase was delayed relative to TG2^+/+^ MEFs (Figure 10A,B). Transfection of TG2^−/−^ MEFs with wild-type TG2 or the addition of wild-type recombinant TG2 to the Fn matrix restored the dynamics of RhoA activation to that of TG2^+/+^ MEFs (*p* = ns comparing time points post-seeding; two-way ANOVA with post hoc Bonferroni correction). GTP-bound Rac1 levels in TG2^−/−^ MEFs increased earlier than in TG2^+/+^ MEFs and plateaued earlier, then fell relative to TG2^+/+^ MEFs (Figure 10C,D). Transfection of TG2^−/−^ MEFs with wild-type TG2 or the addition of wild-type recombinant TG2 to the Fn matrix restored the dynamics of Rac1 activation to that of TG2^+/+^ MEFs. Together, these data indicate alteration of the activation dynamics of RhoA and Rac1, with delayed RhoA activation and earlier Rac1 activation, in 129 TG2^−/−^ MEFs relative to 129 TG2^+/+^ MEFs. These alterations likely explain the delayed cell adhesion and spreading of 129 TG2^−/−^ MEFs.

### 2.11. Closure of an In Vitro Scratch Wound in Confluent Cell Monolayers Was Delayed in 129 but Not B6 TG2^−/−^ MEFs, Relative to Their TG2^+/+^ Counterparts, and the Exogenous Addition of Recombinant Wild-Type TG2 Immediately Post-Wounding Accelerated Wound Closure in All Genotypes

Cell migration into the wound bed is a crucial step in wound healing [1] that can be mimicked by monitoring the healing of in vitro scratch wounds in confluent cell monolayers [47]. Closure of in vitro scratch wounds in confluent monolayers of 129 TG2^−/−^ MEFs was slower than in 129 TG2^+/+^ MEFs (~50% of the day 0 area remaining after 48 h compared to ~10% in TG2^+/+^, Figure 11A,B). Addition of recombinant wild-typeTG2 immediately post-wounding restored motility of 129 TG2^−/−^ MEFs to that of 129 TG2^+/+^ MEFs (*p* = ns comparing time points post-seeding; two-way ANOVA with post hoc Bonferroni correction) and further increased the motility of 129 TG2^+/+^ MEFs, resulting in almost complete wound closure after 48 h. There was no significant difference in in vitro scratch wound healing between B6 TG2^+/+^ and B6 TG2^−/−^ MEFs, and the addition of recombinant wild-typeTG2 increased wound closure to the same extent in both genotypes (Figure 11C,D). These data are consistent with the strain-specific delay observed with in vivo wound healing in 129 TG2^−/−^, but not B6 TG2^−/−^, mice relative to their wild-type counterparts.

### 2.12. TG2 Expression Levels Are Higher in 129 Than in B6 Mice

To determine if the strain-specific wound healing phenotype observed in 129, but not B6 TG2^−/−^, relative to their wild-type counterparts, was due to compensation by other TGs, RNA from unwounded 129 or B6 skin samples and skin samples on day 2 post-wounding were examined for abundance of mRNAs for TG family members (Figure 12A,B). Quantitation revealed no significant differences in *Tgm1*, *Tgm3-7* or *F13A* mRNA levels between 129 and B6 skin of either genotype. Unexpectedly, however, *Tgm2* mRNA abundance was significantly greater in 129 than in B6 TG2^+/+^ skin, both in the unwounded state and post-wounding. Quantitation of TG2 protein abundance in MEFs confirmed significantly greater TG2 abundance in cytosolic and membrane fractions of 129 TG2^+/+^ relative to B6 TG2^+/+^ MEFs (Figure 12C–F). Thus, TG2 abundance in 129 and B6 mouse strains correlates with the rate of wound healing in vivo and MEF wound closure in vitro.

## 3. Discussion

Wound healing is a complex process involving a wide array of cytokines and growth factors that co-ordinate dynamic interactions of multiple cell types with the ECM. Evidence from in vitro cell studies indicates TG2 plays a role in cell adhesion, spreading and motility during wound healing [18,23,29,30,31,32], however, there are no reports of the evaluation of wound healing in TG2^−/−^ mice. Here, we used TG2^−/−^ mice backcrossed to two different background strains, 129 or B6 [33], to investigate the role of TG2 in MEF cellular dynamics in vitro and in wound healing in vivo. We show that in vivo wound healing and in vitro MEF wound closure was delayed in 129, but not B6, TG2 knockouts, relative to their wild-type counterparts, with wound closure in 129 being faster than in B6 wild-types, coincident with higher TG2 expression in 129 than B6 mice. There was no compensation by other TG family members in TG2^−/−^ relative to TG2^+/+^ mice in either strain before or after skin wounding. The addition of TG2 restored the rate of 129 TG2^−/−^ wound healing in vivo and MEF wound closure in vitro to that of 129 TG2^+/+^*,* with in vitro studies confirming this was independent of TG2 transamidase or GTP-binding activity. MEF adhesion was shown to be dependent on syndecan binding in the presence, but not absence, of integrin inhibition. RhoA activation was delayed and Rac1 activation was earlier in 129 TG2^−/−^ than in TG2^+/+^ MEFs. Our findings indicate that although TG2 is not essential for wound closure, it facilitates integrin- and syndecan-mediated RhoA- and Rac1-activation in fibroblasts to promote efficient murine wound contraction in a strain-dependent manner.

Impaired wound healing of 129 TG2^−/−^ relative to 129 TG2^+/+^ mice was observed during the inflammatory and early cell proliferation phases of wound healing (Figure 1). Unlike knockout models of MMP8 [48] and PPARα [49], which have attributed delayed early phase wound healing to delayed neutrophil, or neutrophil and monocyte infiltration, respectively, during the first few days post-wounding, impaired wound healing in 129 TG2^−/−^ mice was not associated with an impairment of neutrophil or monocyte recruitment to the wound bed (Figure 3). Rather, TG2 expression was induced at this time in 129 TG2^+/+^ mice as part of the ’wounded’ transcriptome (Figure 4). This observation is consistent with in vitro studies, which have shown that TG2 expression is induced by thrombin and cytokines, such as TNFα, INF-γ and TGF-β [50,51,52,53], all of which are released during early wound healing (Figure 4) by inflammatory cells [1]. Included in the ’wounded’ ECM transcriptome was increased abundance of the wound-related fibronectin extra-domain A splice variant (Figure 4), which is induced by TGF-β in fibroblasts, macrophages and endothelial cells during cutaneous wound healing [54,55,56] and which promotes fibroblast adhesion and spreading [57] via the same integrins, α9β1 and α4β1 [58], with which TG2 interacts [59]. The acceleration of 129 TG2^−/−^ cutaneous wound healing (Figure 5) and of 129 TG2^−/−^ and TG2^+/+^ scratch wound closure (Figure 11) that was observed upon the addition of wild-type TG2 immediately post-wounding, indicates an early and direct interaction of TG2 protein within the wound bed to facilitate wound closure. Given that increased MEF adhesion capacity and acceleration of MEF spreading was observed upon the addition of wild-type, transamidase-defective or GTP-binding-defective TG2 (Figure 6, Figure 7 and Figure 9), this interaction of TG2 is transamidase- and GTP-binding-independent and is localised to the β-sandwich-core domain of TG2 (Appendix A). The markedly delayed wound healing observed in both 129 TG2^−/−^ and 129 TG2^+/+^ mice upon local site administration of bupivacaine (Figure 2) likely reflects the inhibition of cytokine production by inflammatory cells [35] in the inflammatory phase of wound healing and, consequently, abrogation of the wound healing advantage that TG2 provides to 129 TG2^+/+^ mice relative to 129 TG2^−/−^ mice. Thus, although not essential to wound healing, the environment of the wound enables promotion of wound healing by TG2.

Fibroblast adhesion to Fn is controlled synergistically by integrins and syndecans [3]. The cytoplasmic domains of integrins interact with the actin cytoskeleton and syndecans directly regulate protein kinase α [3]. A body of literature details TG2 as a co-receptor for integrin and syndecan-4 [18,19,20,21]. In support of this, we observed that the adhesion capacity of 129 TG2^−/−^ MEFs was lower than that of 129 TG2^+/+^ MEFs (Figure 6, Figure 7 and Figure 8). Adhesion involved both integrins and syndecans, with adhesion being dependent on syndecan binding in the presence, but not absence, of RGD inhibition of integrin binding (Figure 8). The Rho family of GTPases are central to cytoskeletal re-organization during cell adhesion and spreading, the best characterized being RhoA and Rac1. RhoA stimulates the formation of actin stress fibers and large focal adhesions, while Rac1 induces cell spreading through the formation of smaller focal complexes at the plasma membrane and membrane ruffles or lamellipodia [44,45,46]. Our observation that RhoA and Rac1 activation were both altered, with RhoA activation being delayed and Rac1 activation being earlier in 129 TG2^−/−^ relative to 129 TG2^+/+^ MEFs (Figure 10), is consistent with the activation dynamics of RhoA and Rac1 in syndecan-4 knockout MEFs [60], which also exhibit a delay in scratch wound closure in vitro that is reflected in delayed cutaneous wound healing in vivo [61]. Thus, our data suggest that TG2 facilitation of murine wound contraction involves integrin- and syndecan-mediated RhoA- and Rac1-activation.

The delayed cellular dynamics and wound healing phenotype in the TG2 knockout mice was strain dependent and evident only on the 129, and not the B6, background (Figure 1, Figure 6, Figure 9 and Figure 11). TG2 was more abundant in the cytosol and membrane of 129 MEFs than in B6 MEFs (Figure 12), which likely explains the faster wound closure of wild-type 129 relative to wild-type B6 (Figure 1 and Figure 11). The increase in wild-type 129 MEF adhesion capacity, spreading and wound closure observed upon addition of TG2 (Figure 7, Figure 8, Figure 9 and Figure 11) suggests that the amount of TG2 in 129 cells is not saturating with respect to other factor(s) that modulate the cellular dynamics of wound contraction. Further work will be required to delineate additional strain-dependent factor(s) that underlie the observed phenotypic difference as well as the relevance of TG2 to human skin wound healing.

In conclusion, our findings show that the rate of wound healing between 129 and B6 mouse strains is different and directly related to differences in TG2 abundance, and although not essential for wound healing, TG2 facilitates integrin- and syndecan-mediated RhoA- and Rac1-activation in fibroblasts to promote efficient murine wound contraction.

## 4. Materials and Methods

### 4.1. Ethics Statement

All experimental procedures were approved by the Garvan Institute/St. Vincent’s Hospital Animal Experimentation Ethics Committee (No. 07/12, 09/30) and were performed in strict accordance with the National Health and Medical Research Council (NHMRC) of Australia Guidelines on Animal Experimentation. All efforts were made to minimize suffering. All analyses were performed with the operator blinded to genotype.

### 4.2. Animal and Cell Culture

Heterozygous TG2 knockout mice (Tgm2^tm1.1Rmgr^) [62], congenic on either a C57BL/6J (B6) or 129T2/SvEmsJ (129) background [33], were maintained and mated in-house under specific pathogen-free conditions to generate TG2 wild-type (TG2^+/+^) or knockout (TG2^−/−^) breeding pairs for generation of TG2^+/+^ and TG2^−/−^ mice for experimentation. C57BL/6J and 129T2/SvEmsJ were originally obtained from the Jackson Laboratory (Bar Harbor, ME, USA).

### 4.3. Recombinant Rat TG2 Production

Wild-type TG2, domain deletion mutants of TG2 (β-sandwich, core, β-sandwich-core) [41], the transamidase-deficient point mutant W241A [63], and the GTP-binding mutant R579A [40] were expressed as glutathione-S-transferase-TG2 fusion proteins in pGEX-2T expression vector (GE Healthcare, Sydney, NSW, Australia) in *Escherichia coli* M15 and purified as described previously [63]. TG2 used for animal experiments was dialyzed (6–8000 molecular weight cut-off) against sterile PBS (12 h, 4 °C). The protein was concentrated to 10 mg/mL with Ultragel 30K cut-off (Millipore, Burlington, MA, USA) and stored at 4 °C until use.

### 4.4. In-Vivo Wound Healing Assay

129 or B6 TG2^+/+^ or TG2^−/−^ male mice aged 12 ± 2 weeks (*n* = 120) had their backs gently shaved with an electric razor 1 day prior to assay and animal identities were blinded. On the day of surgery, mice were anesthetized with 2.5% isofluorane/oxygen. Sterile 0.9% saline or bupivacaine (8 mg/kg) was injected subcutaneously 30 mm from the neck and 10 mm from either side of the spine to create a bulge on which two circular dorsal full thickness excisional wounds were made using a punch biopsy tool (Stiefel Laboratory, Research Triangle Park, NC, USA). Wounds were left uncovered. Mice were allowed to recover housed singly in dust-free bedded cages and were monitored closely over the first 8 h and for 3 days post-wounding for any signs of pain, coat condition/grooming, changes in food intake, alertness or mobility; none were evident. In some experiments, mice were re-anesthetized with 2.5% isofluorane/oxygen at 30 min post-surgery for application of purified TG2 protein (50 μL of 10 mg/mL in PBS) to one wound (side chosen by the flip of a coin) and PBS (50 μL) as a control to the other wound; mice were maintained under 1% isofluorane until the solution was absorbed into the wound bed. Wound closure was monitored daily under 2.5% isofluorane/oxygen sedation by digital photography using a Canon IXUS 50 camera (Canon, Tokyo, Japan) at a fixed perpendicular distance on a retort stand from day 0. Epidermal wound edges were also traced onto a sterile coverslip. The wound area (expressed as fraction of day 0 wound size) was measured using the software ImageJ v1.46c. The integrated wound closure time was calculated (total area under the curve of wound area fractions over time) to give an indication of the total time that the wound was exposed.

### 4.5. Histological Analysis of Cutaneous Wounds

On each day post-wounding, up to day 5, mice (*n* = 3) were culled by cervical dislocation and wounds were quickly isolated by cutting 2 mm around the wound’s outer edge. Liberated wounds were gently lifted using sterile fine forceps and connective tissue and/or fat were removed using sterile surgical scissors. Wounds were fixed in 4% paraformaldehyde in PBS overnight at 4 °C. Fixed wounds were washed and then submerged in 70% ethanol overnight at room temperature. Samples were subsequently embedded in paraffin wax blocks and sectioned transversely to produce 6 μm thick sections. The sections were mounted onto a glass slide and stained with hematoxylin and eosin for monocytes and neutrophils. Areas in the immediate vicinity of the wound’s edge were examined under a Leica DMRF fluorescent microscope (Leica Microsystems GmBH, Wetzlar, Germany); sections were visualized at 20× magnification. For each sample up to day 5 (*n* = 3), ten random fields were viewed, and monocytes and neutrophils were counted. Monocytes were identified as cells with a blue-stained bean-shaped nucleus and neutrophils were identified as cells with a blue-stained multi-lobed nucleus, with thin strands connecting each lobe.

### 4.6. Wound RNA Extraction

Male 129 TG2^+/+^ and TG2^−/−^ mice were subjected to wound healing assay and on day 2 post-injury, mice (*n* = 6) were culled, wounds were isolated by cutting about 2 mm outside of the wound’s outer edge as described above in ’Histological analysis of cutaneous wounds’ and quickly placed into a microfuge tube, on ice, containing 700 μL of QIAzol Lysis Reagent from miRNeasy RNA extraction kit (QIAGEN, Hilden, Germany). The tissue was homogenized on ice by PRO 200 homogenizer (PRO Scientific, Oxford, CT, USA) until minced. RNA extraction was continued following the manufacturer’s instructions. Purified RNA pellets were dissolved in 50 μL of RNase-free water at room temperature or at 60 °C for 10 min. RNA concentration was measured using a NanoDrop 2000 spectrometer (Thermo Fisher Scientific, Sydney, NSW, Australia) and subsequently stored at −80 °C in 200 ng aliquots. An aliquot of RNA from each sample was measured for RNA integrity using Agilent Bioanalzyer 2100 (Agilent Technologies, Santa Clara, CA, USA). Samples were loaded onto an RNA 6000 Nano Chip (Agilent Technologies, Santa Clara, CA, USA) according to the manufacturer’s instructions. If the cutaneous sample RNA integrity number was above 7.0, it was deemed good quality and used for subsequent microarray analysis.

### 4.7. Microarray Analysis of Total RNA Isolated from Cutaneous Wounds

A total of 500 nanograms of total RNA were processed at the Ramaciotti Centre for Genomics (UNSW, Sydney, NSW, Australia) using the Agilent QuickAmp Labeling Kit, One colour (PN 5190-0442; Agilent Technologies, Santa Clara, CA, USA) and the 3′ Gene Expression Hybridization kit (PN 5188-5242; Agilent Technologies, Santa Clara, CA, USA). Labelled RNA was hybridized to a one-coloured GeneChip Mouse Exon 1.0 ST Array (Affymetrix, Sunnyvale, CA, USA), which contains 1.2 million probes, each corresponding to one unique genomic element in the mouse genome. Quality controls included internal grid controls and RNA spike-ins. The array had control spots to varied amounts of prokaryotic genes from the biotin synthesis pathway: *bioB*, *bioC* and *bioD* from *E. coli* and Cre recombinase from P1 bacteriophage, and each sample was spiked with the complementary RNA for these genes. Array scanning and subsequent data extraction were performed by the Ramaciotti Centre and the Peter Wills Bioinformatics Centre (Garvan Institute for Medical Research, Sydney, NSW, Australia). Microarray data were analyzed using the software GeneSpring vGX 11 (Agilent Technologies, Santa Clara, CA, USA) for all genes meeting the threshold values of a relative log-fold-change of at least 0.5 and *p* < 0.05.

### 4.8. Quantitative PCR (qPCR) Analysis of Wound RNA

Complementary DNA was synthesized from the total RNA isolated from wounds by first heating 11 μL of total RNA with 2 μL of oligo(dT)20 primers for 5 min at 65 °C followed by addition of 1 mM of deoxynucleotide triphosphate (Invitrogen, Carlsbad, CA, USA), 1× FS buffer (Invitrogen), 0.05U RNAsin (Promega, Madison, WI, USA), and 200U Superscript III reverse transcriptase to a total volume of 20 μL and incubation at room temperature for 5 min, 50 °C for 60 min, and 75 °C for 15 min. The resultant complementary DNA was diluted to 50 μL. Real-time quantitative PCR was performed on cDNA prepared from RNA isolated from TG2^+/+^ or TG2^−/−^ mouse wounds (day 2; *n* = 3) or cDNA prepared from RNA isolated from healthy skins (*n* = 3) using Taqman Gene Expression Assay 384 well format (Applied Biosystems, Waltham, MA, USA). Quantitative PCR was performed in triplicate to examine the expression levels of *Tgm1* (assay ID: Mm00498375_m1*), *Tgm2* (Mm00436987_m1*), *Tgm3* (Mm00436999_m1*), *F13a1* (Mm00472334_m1*), *Tgm4* (Mm00626039_m1*), *Tgm5* (Mm00551325_m1*), *Tgm6* (Mm00624922_m1*), *Tgm7* (Mm03990491_m1*), *Il6* (Mm00446190_m1*), *Ifng* (Mm00497611_m1*), Tnf (Mm00443258_m1*), Egf (Mm00438696_m1*), Fgf2 (Mm00433287_m1*), *Tgfb2* (Mm00436955_m1*), *Ticam1* (Mm00844508_s1*), *Myd88* (Mm00440338_m1*), *Nfkb1* (Mm00476361_m1*), *Nfkbib* (Mm00456849_m1*), *Fndc4* (Mm00480765_m1*), *Fndc3a* (Mm01232694_m1*), *Col1a1* (Mm00801666_g1*), *Col1a2* (Mm00483888_m1*), *Col3a1* (Mm01254476_m1*), *Sdc4* (Mm00488527_m1*), with *Hprt* (Mm00446968_m1*) as the most suitable reference gene of four reference genes (*Hprt*, *GAPDH*, 18S rRNA and β-2-microglubin (*B2M*)) tested (expression unaffected by the experimental treatment). Each reaction was performed in a 10 μL reaction mix: 5 μL LightCycler probe master mix (Applied Bioscience, Wilmington, NC, USA); 0.5 μL Taqman probes; 2.5 μL water; and 2 μL of cDNA. A standard curve using TG2^+/+^ cDNA dilutions (1, 1:10, 1:100, and 1:1000) gave similar PCR amplification efficiencies for the reference and target genes (90–110%), thereby validating the use of the 2^−ΔΔCT^ method for each gene examined. A no template control reaction was included for each gene examined. The 384-well plate was sealed and the reaction was incubated in a LightCycler 480 (Roche, Basel, Switzerland) with the melting temperature at 60 °C for all probe sets. Results were only accepted if the crossing point was between 20–30 cycles and if the standard deviation within one triplicate was less than 0.5.

### 4.9. Isolation and Maintenance of Mouse Embryonic Fibroblasts

Homozygous female mice were time-mated with males of the same genotype. Following evidence of a copulatory plug, the female was weighed and separated. At 13.5 days post-coitus, the impregnated female was sacrificed by cervical dislocation (the weight at this stage is ~1.4 × the mother’s weight at day 0 for a mother carrying six embryos). Using sterile surgical scissors, an incision was made in the abdominal cavity and the embryo-containing uterus was excised into a petri dish containing sterile PBS, pH 7.4 (0.13 M NaCl; 2.6 mM KCl; 10 mM Na_2_HPO_4_; 1.7 mM KH_2_PO_4_). In a class II biological safety cabinet (AES Environmental, Adelaide, SA, Australia), the placenta, uterus wall, Reichert’s membrane and visceral yolk sac were removed to expose individual embryos. The head was removed from each embryo and the remaining body was incubated in 1× trypsin-EDTA solution (Sigma-Aldrich, Saint Louis, MI, USA; 1 mL/embryo, 16 h, 4 °C) in a 15 mL Falcon tube (BD Biosciences, San Jose, CA, USA). Embryos were subsequently digested at 37 °C for 15 min and minced by gentle trituration using a 25 mL pipette followed by a 10 mL pipette. The trypsin-cell solution was neutralized with 5 mL DMEM/10% FCS (Dulbecco’s modified Eagles’ medium high glucose (Invitrogen, Carlsbad, CA, USA); 10% *v*/*v* fetal calf serum (Invitrogen); 400 μM L-Glutamine (Invitrogen); 0.2 U/mL penicillin (Invitrogen); 0.2 mg/mL Streptomycin (Invitrogen)) at 37 °C and centrifuged (120× *g*, 5 min). Cells were resuspended in DMEM/10% FCS at 1 mL/embryo for plating onto 15 cm tissue culture dishes (1 mL/plate) containing 15 mL DMEM/10% FCS, incubated at 37 °C under 5% CO_2_ and marked as passage one. At confluency, the MEF culture was cleared of any large embryo debris by vacuum suction with a Pasteur pipette. Adherent cells were detached by incubation with 1× trypsin-EDTA solution (37 °C, 5 min). Detached cells were confirmed by examination under a Nikon Eclipse TS100 microscope (Nikon, Tokyo, Japan) for floating luminescent spheroid cells. Trypsin was neutralized by addition of at least 1 × volume DMEM/10% FCS. Cells were collected by centrifugation (120× *g*, 5 min) and stored in liquid N_2_ or maintained from passages 2 through to 5 on 15 cm tissue culture dishes with DMEM/10% FCS.

### 4.10. Cell Proliferation Assay

On consecutive days post-seeding, cells from each passage were detached with Trypsin-EDTA and total live cell counts were determined by counting in a Coulter counter with a gating range of 7 μm to 24 μm diameter.

### 4.11. In Vitro Cell Adhesion Assay

In total 24-well tissue culture plate wells were incubated with bovine fibronectin (10 μg/cm^2^, pH 7.4, 15 h, 4 °C) and then non-specific binding sites blocked (3% BSA in PBS) for non-specific binding sites. To some pre-coated wells, purified recombinant rat TG2 (wild-type, W241A-TG2 or R579A-TG2 at 10, 20, 40, 60 or 80 μg/cm^2^) was added and incubated for 30 min at 37 °C. Sub-confluent 129 or B6 TG2^+/+^ or TG2^−/−^ MEFs (passages 2–5) were synchronised in DMEM/0.1% FCS (supplemented with 100 U/mL penicillin, 10 μg/mL streptomycin) for about 15 h. Fibroblasts were detached using enzyme-free dissociation solution (Millipore, Burlington, MA, USA), and seeded at 5 × 10^4^ cells/cm^2^ onto treated wells in DMEM/0.1% FCS. In some wells, heparin sulphate (42 μg/5 × 10^4^ cells) and/or GRGDTP or GRADSP peptides (both 84 μg/5 × 10^4^ cells) were added for the inhibition of adhesion pathway(s). Seeded MEFs were incubated at 37 °C under 5% CO_2_. Adherent cells were washed once with cold PBS, fixed with 4% paraformaldehyde in PBS (15 min, room temperature), stained with 0.1% crystal violet stain in 25% methanol (15 min, room temperature) and washed three times with PBS. Crystal violet was eluted with 200 μL 70% ethanol and stain intensity was read at 600 nm to quantitate adherent cells. Separate negative controls without MEFs as well as an internal standard (constructed by seeding cells and then cross-linking them to the pre-coated dishes by incubation with paraformaldehyde) were included for each treatment.

### 4.12. In Vitro Cell Spreading Assay

Synchronized 129 or B6 TG2^+/+^ or TG2^−/−^ MEFs were seeded onto pre-treated sterile coverslips, incubated, washed and fixed as described above in ’In vitro cell adhesion assay’. Adhered cells were permeabilized (0.1% Triton-X/3% BSA in PBS, room temperature, 15 min), washed with PBS and then stained for actin filaments with 70 nM phalloidin-TRITC in PBS. Coverslip samples were mounted onto Super Frost Slides using Prolong Gold Antifade with DAPI mounting medium (Invitrogen). Samples were laser-excited at 405 nm (DAPI) and 555 nm (TRITC) to detect nuclei and cytosolic actin, respectively, and ten random fields (20× magnification) were captured with an LSM 700 microscope (Zeiss, Oberkochen, Germany). Cell area was evaluated using the software ImageJ v1.46c, and only non-overlapping mono-nucleated cells were analyzed.

### 4.13. Transfection of MEFs with TG2-IRES-GFP Expression Vectors

Expression constructs were generated by first sub-cloning cDNA encoding wild-type TG2 or various point mutants (transamidase-deficient: W241A, C277; GTP-binding-deficient: R579A, S171E) into pcDNA3.1 as described previously [40]. Plasmids were linearized by a *Not*I digestion (2 h, 37 °C), 3′-recess end-filled using large Klenow fragment (1 h, 37 °C), followed by *Eco*RI digestion (2 h, 37 °C). The resulting 2.1 kb DNAs encoding TG2 were cloned into *Sma*I–*Eco*RI sites of pIRES-EGFP (Clontech, Mountain View, CA, USA). Transfection of 129 TG2^−/−^ MEFs with wild-type or mutant TG2-IRES-GFP plasmids was performed using the Amaxa nucleofection system (Lonza Group Ltd., Basel, Switzerland) in accordance with the manufacturer’s instructions. Visual inspection for green fluorescent protein fluorescence was confirmed under a LSM 700 laser scanning microscope (Zeiss, Oberkochen, Germany) at an excitation wavelength of 488 nm. As the GFP sequence is expressed at reduced efficiency from the IRES, with respect to the upstream cloned TG2 sequence, GFP-positive cells were taken to be TG2-positive as well. The transfection efficiency (32 ± 4.3%) was determined on a proportion of transfected cells by flow analysis and gating of green fluorescent protein (GFP)-positive cells (488 nm ex) using the FACSCanto II flow cytometer (BD Biosciences, San Jose, CA, USA). Wild-type and mutant protein expression was equalized by varying the amount of plasmid construct transfected (wild-type TG2, 10 μg/2 × 10^6^ cells; W241A, 9.6 μg/2 × 10^6^ cells; C277S, 11.5 μg/2 × 10^6^ cells; R579A, 10 μg/2 × 10^6^ cells; S171E, 11.5 μg/2 × 10^6^ cells). Equivalent TG2 protein expression was confirmed in a proportion of transfected cells from each transfection experiment using FACS for GFP-positive cells, followed by cell lysis and Western blot analysis for TG2 protein expression in membrane and cytoplasmic fractions, normalized to pan-cadherin and GAPDH expression, respectively. Untreated 129 TG2^−/−^ MEFs and 129 TG2^−/−^ MEFs transfected with empty pIRES-EGFP were included as negative controls. Transfected MEFs were allowed to recover for 24 h before further experimentation.

### 4.14. RhoA and Rac1 Profiling in MEFs

*E. coli* Rosetta haminrboring either the GST-Rhotekin Rho-binding domain (RBD) (gift from Martin Schwartz (Addgene plasmid #15247, http://n2t.net/addgene:15247 (accessed on 30 June 2023), RRID:Addgene_15247) [44]) or GST-p21 activated kinase 1 protein (PAK) Rac/Cdc42 (p21)-binding domain (PBD) [64], both in pGEX2T, were induced to express GST-Rhotekin-RBD or GST-PAK-PBD overnight at 37 °C with 1 mM IPTG in 2 × YT medium containing ampicillin (250 μg/mL). After sonication in PBS and centrifugation (16,100× *g*, 10 min, room temperature), the supernatant fraction was loaded onto PBS-pre-washed glutathione sepharose beads (45 min, 4 °C). Beads were resuspended in PBS (600 μL) and a 1 μL sample was quantitated against a BSA standard curve developed with Coomassie plus reagent (Abs 600 nm). Protein-bound beads were divided into 50 μg aliquots in 100 μL of GST-Fish buffer (10% *v*/*v* glycerol, 50 mM Tris-HCl buffer pH 7.4, 100 mM NaCl, 1% Tergitol-type NP-40, and 2 mM MgCl_2_). Serum-starved MEFs in DMEM/0.1% FCS (1 × 10^6^ cells) were allowed to settle on plates coated with Fn (10 μg/cm^2^) or Fn with TG2 (20 μg/cm^2^) added for 10, 30, 60 or 90 min. At these time points, 129 TG2^+/+^ or TG2^−/−^ MEFs were placed on ice, media was aspirated and cells were scraped in 500 μL of GST-Fish buffer using a rubber policeman. Cells were lysed by trituration, transferred to a fresh microfuge tube and centrifuged (16,100× *g*, 5 min). A portion of the lysate (100 μL, pre-pull-down) was transferred to a new tube containing six x Laemmli buffer, and the remainder was added to 50 μg of either GST-Rhotekin-RBD or GST-Pak-RBD and incubated for 1 h with constant rotation. Incubated beads were collected by centrifugation (1000× *g*, 5 min) and washed twice with GST-Fish buffer. After the last centrifugation, the supernatant was discarded and 25 μL of GST-Fish plus 5 μL of Laemmli buffer were added. Both bead samples and pre-pull-down lysates were boiled (100 °C, 5 min) and size-fractionated on a 15% reducing polyacrylamide gel (15% acrylamide/bis solution, 37.5:1, 2.6% cross-linker monomer (Bio-Rad, Sydney, NSW, Australia); 0.37M Tris, pH 8.8, 0.01% SDS). Samples were subjected to Western blotting using rabbit anti-mouse RhoA polyclonal antibody (ab86297; Abcam, Cambridge, UK) to detect RhoA and rabbit anti-mouse Rac1 polyclonal antibody (Abcam, ab155938) to detect Rac1. A positive control was included where TG2^+/+^ MEF lysate was incubated with 5 mM GTPγS (30 min, 37 °C) prior to pull-down with beads. Bands shown are from exposures that optimize visualization of the target protein (e.g., GTP-RhoA). Separately, the same gels were also exposed for shorter times (so as not to exceed Beer’s Law), to allow accurate quantitation of the total protein (e.g., total RhoA), and these were used to normalize the target protein results.

### 4.15. In Vitro Scratch Wound Assay

129 or B6 TG2^+/+^ or TG2^−/−^ MEFs were grown to confluency in six-well plates in DMEM/10% FCS. A scratch was made in the fibroblast monolayer using a sterile 200 μL tip. The border of the denuded area was immediately marked under a Nikon eclipse TS100 microscope at 10× magnification. The denuded area was recorded at 0, 24 and 48 h post-scratching. In some experiments, purified TG2 (500 μg/mL) was added immediately to wells containing TG2^−/−^ MEFs. Denuded areas were evaluated using the software ImageJ and expressed as a fraction of day zero area.

### 4.16. Western Blotting

MEFs were lysed in RIPA buffer (500 μL of 50 mM Tris-HCl, pH 7.4; 150 mM NaCl; 0.1% sodium dodecyl sulphate; 0.5% sodium deoxycholate; 1% Tergitol-type NP-40 (Sigma-Aldrich, Sydney, NSW, Australia) with fresh inhibitor cocktail (1 mM phenylmethanesulfonylfluoride; 0.4 mM Aprotinin; 0.4 mM Leupeptin; 0.4 mM Pepstatin) per 10 cm dish), passed through a 1 mL insulin syringe 3 times, sonicated (Branson 400 Digital Sonifier on ice, 40% amplitude output, 2 × 2 min with a 2 min break on ice in between each sonication) and centrifuged in a microfuge tube (1000× *g*, 20 min, 4 °C). The supernatant was transferred to a new microfuge tube and centrifuged (16,100× *g*, 1 h, 4 °C). The supernatant was the cytoplasmic fraction and the pellet (resuspended in 50 μL of RIPA buffer) was the membrane fraction. Both fractions were size-fractionated on an 8% reducing polyacrylamide gel (8% acrylamide/bis solution, 37.5:1, 2.6% cross-linker monomer [Bio-Rad]; 0.37M Tris, pH 8.8, 0.01% SDS) and subjected to Western blotting using rabbit anti-mouse TG2 rat monoclonal antibody (a gift from Gail V.W. Johnson, University of Rochester, Rochester, NY, USA; 1:16,000 dilution in Tris-buffered saline, 1 h, room temperature). Rabbit anti-mouse GAPDH polyclonal antibody (1:5000; Abcam) and rabbit anti-mouse pan-Cadherin polyclonal antibody (1:2500; Abcam) served as the most suitable reference protein for loading (expression unaffected by genotype) for cytoplasmic and membrane fractions, respectively. Blots were developed with enhanced chemiluminescence solution (GE Healthcare) as per the manufacturer’s instructions. Expression was measured by densitometric analysis using ImageJ software v1.46c.

### 4.17. Statistical Analysis

All results are shown as mean ± standard error of the mean (SEM). Statistical analyses were performed using GraphPad Prism v9 (Graphpad Software Inc, La Jolla, CA, USA). A two-tailed Student’s *t* test was used for comparisons between two groups, and one-way ANOVA with Bonferroni’s multiple group comparisons or two-way ANOVA with post hoc Bonferroni correction was used for multiple comparisons. In all cases where Student’s *t* test was used, the data passed the Shapiro–Wilk normality test (*p* = ns), which assesses normality with standard deviation, thereby indicating the data are not inconsistent with a Gaussian distribution. Statistical significance was considered to have been reached with a *p* value < 0.05.

## Figures and Tables

**Figure 1 ijms-24-11475-f001:**
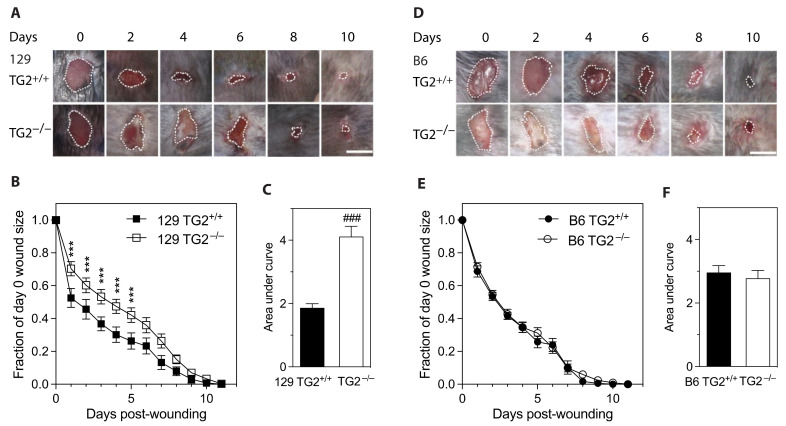
Early healing of cutaneous wounds was slower in 129 TG2^−/−^ mice than in 129 TG2^+/+^, B6 TG2^+/+^ or B6 TG2^−/−^ mice. (**A**,**D**) Representative photographs of cutaneous wounds in (**A**) 129 TG2^+/+^ or 129 TG2^−/−^ or (**D**) B6 TG2^+/+^ or B6 TG2^−/−^ mice at 0, 2, 4, 6, 8 and 10 days post-wounding. (**B**,**E**) Wound areas expressed as a fraction of the day 0 area of (**B**) 129 TG2^+/+^ (*n* = 16) or 129 TG2^−/−^ (*n* = 16), respectively, or (**E**) B6 TG2^+/+^ (*n* = 10) or B6 TG2^−/−^ (*n* = 10), respectively, until wound closure. (**C**,**F**) Integrated wound closure time: (**C**) calculated from total area under the curves from (**B**); (**F**) calculated from total area under the curves from (**E**). Scale bar = 5 mm. ***, *p* < 0.001 for individual post hoc Bonferroni test results between TG2^−/−^ and TG2^+/+^ from repeated measure two-way ANOVA; ###, *p* < 0.001 for two tailed Student’s *t* test.

**Figure 2 ijms-24-11475-f002:**
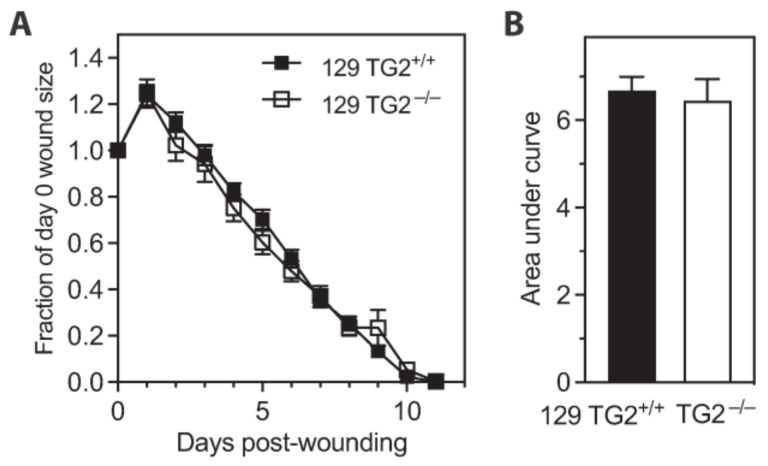
Bupivacaine injection prior to skin punch biopsy delayed wound healing in 129 TG2^−/−^ and 129 TG2^+/+^mice. (**A**) Wound areas expressed as a fraction of the day 0 area of 129 TG2^+/+^ (*n* = 16) or 129 TG2^−/−^ (*n* = 16), respectively, until wound closure. (**B**) Integrated wound closure time calculated from total area under the curves from (**A**).

**Figure 3 ijms-24-11475-f003:**
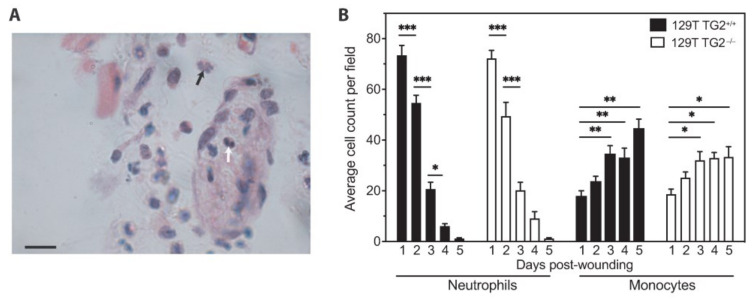
Neutrophil and monocyte recruitment to cutaneous wounds was no different between 129 TG2^−/−^ and 129 TG2^+/+^. (**A**) A typical random field (60× magnification) of a transverse cutaneous wound section showing an example of a multi-lobed nucleus of a neutrophil (black arrow) and a bean-shaped nucleus of a monocyte (white arrow); scale bar = 10 μm. (**B**) Number of neutrophils and monocytes per field (average of 10 random 20× magnification fields from *n* = 3 samples per genotype) in transverse sections of 129 TG2^+/+^ or 129 TG2^−/−^ cutaneous wounds 1–5 days post-wounding. *, *p* < 0.05; **, *p* < 0.01, ***, *p* < 0.001 for individual post hoc Bonferroni test results from two-way ANOVA.

**Figure 4 ijms-24-11475-f004:**
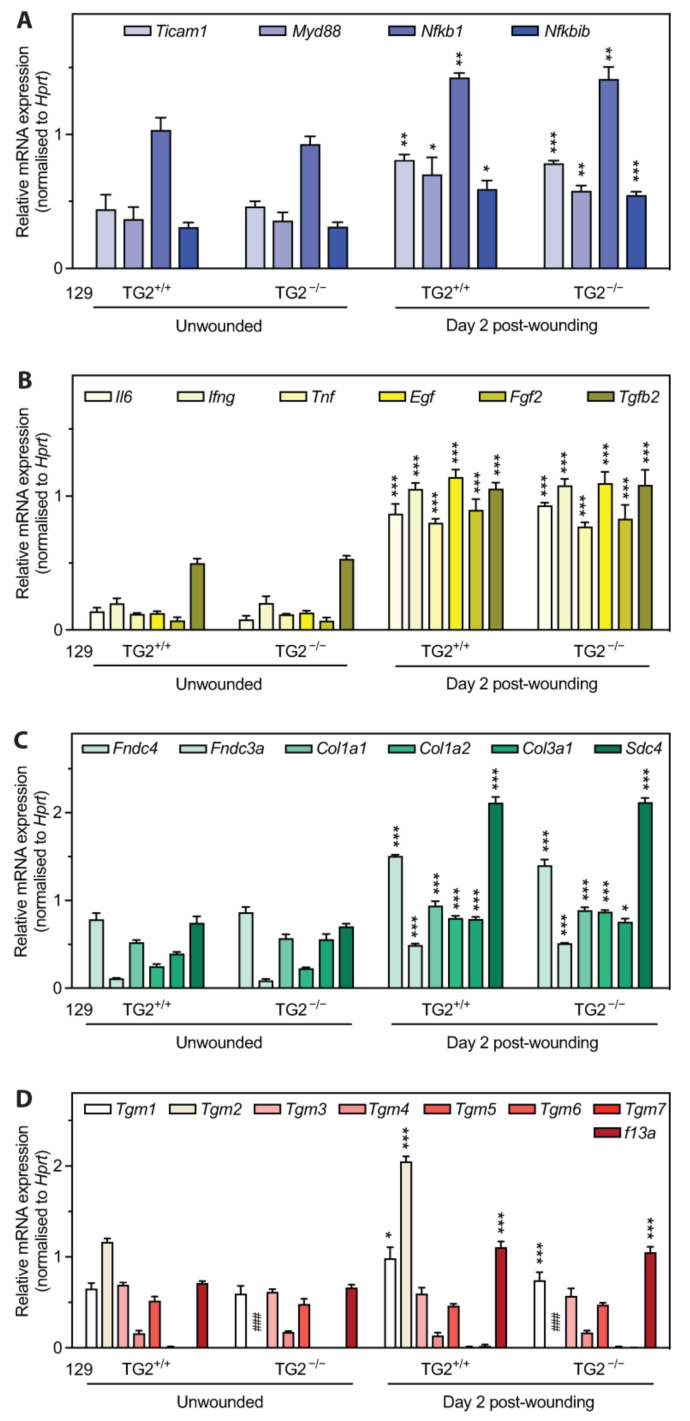
mRNA abundance of wound-related genes was increased to the same levels in 129 TG2^−/−^ and 129 TG2^+/+^ mice on day 2 post-wounding. cDNA generated from total RNA isolated from unwounded skin samples (*n* = 3 per genotype) or skin samples on day 2 post-wounding (*n* = 3 per genotype) was examined in triplicate for the abundance of mRNAs for (**A**) TLR pathway genes *Ticam1* (encoding TICAM1), *Myd88* (encoding MyD88), *Nfkb1* (encoding NF-κB subunit 1), *Nfkbib* (encoding I-κBβ), (**B**) early inflammatory cytokines *Il6* (encoding IL6), *Ifng* (encoding IFN-γ) and *Tnf* (encoding TNF-α), and growth factors *Egf* (encoding EGF), *Fgf2* (encoding basic fibroblast growth factor) and *Tgfb2* (encoding TGFβ2), (**C**) ECM-related genes *Fndc4* (encoding Fibronectin Type III Domain-Containing Protein 4), *Fndc3a* (encoding the wound-related fibronectin extra-domain A splice variant), *Col1a1* (encoding collagen type I α1 chain), *Col1a2* (encoding collagen type I α2 chain), *Col3a1* (encoding collagen type III α1 chain), *Sdc4* (encoding syndecan-4), (**D**) TG family members *Tgm1* (encoding TG1), *Tgm2* (encoding TG2), *Tgm3* (encoding TG3), *Tgm4* (encoding TG4), *Tgm5* (encoding TG5), *Tgm6* (encoding TG6), *Tgm7* (encoding TG7), *f13a* (encoding F13A), normalized to *Hprt* (encoding hypoxanthine phosphoribosyltransferase 1) as the reference gene (expression level unaffected by the experimental treatment). *, *p* < 0.05; **, *p* < 0.01; ***, *p* < 0.001 for two tailed Student’s *t* test between respective different treatment groups of the same genotype. ###, *p* < 0.001 for two tailed Student’s *t* test between different genotypes in the same respective treatment group.

**Figure 5 ijms-24-11475-f005:**
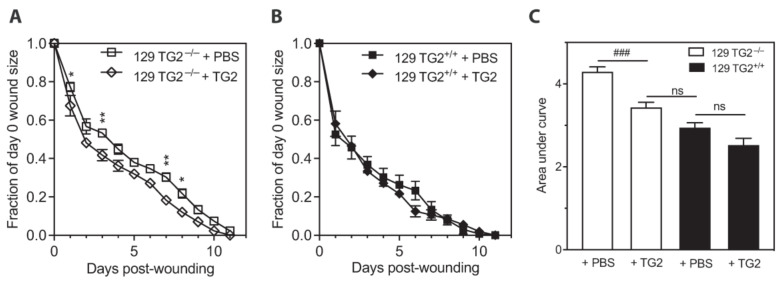
A single dose of TG2 protein to cutaneous wounds in 129 TG2^−/−^ mice restored rate of wound healing to that in 129 TG2^+/+^ mice. (**A**,**B**) Wound areas expressed as a fraction of the day 0 area of (**A**) 129 TG2^−/−^ wounds (*n* = 9) or (**B**) 129 TG2^+/+^ wounds (*n* = 9) treated with vehicle (PBS) or TG2 until wound closure. (**C**) Integrated wound closure time calculated from the total area under the curves from (**A**,**B**). *, *p* < 0.05; **, *p* < 0.01 for individual post hoc Bonferroni test results between TG2^−/−^ + PBS and TG2^−/−^ + TG2 from repeated measure two-way ANOVA with post hoc Bonferroni correction. ###, *p* < 0.001; ns, not significant for one-way ANOVA with Bonferroni’s multiple group comparisons.

**Figure 6 ijms-24-11475-f006:**
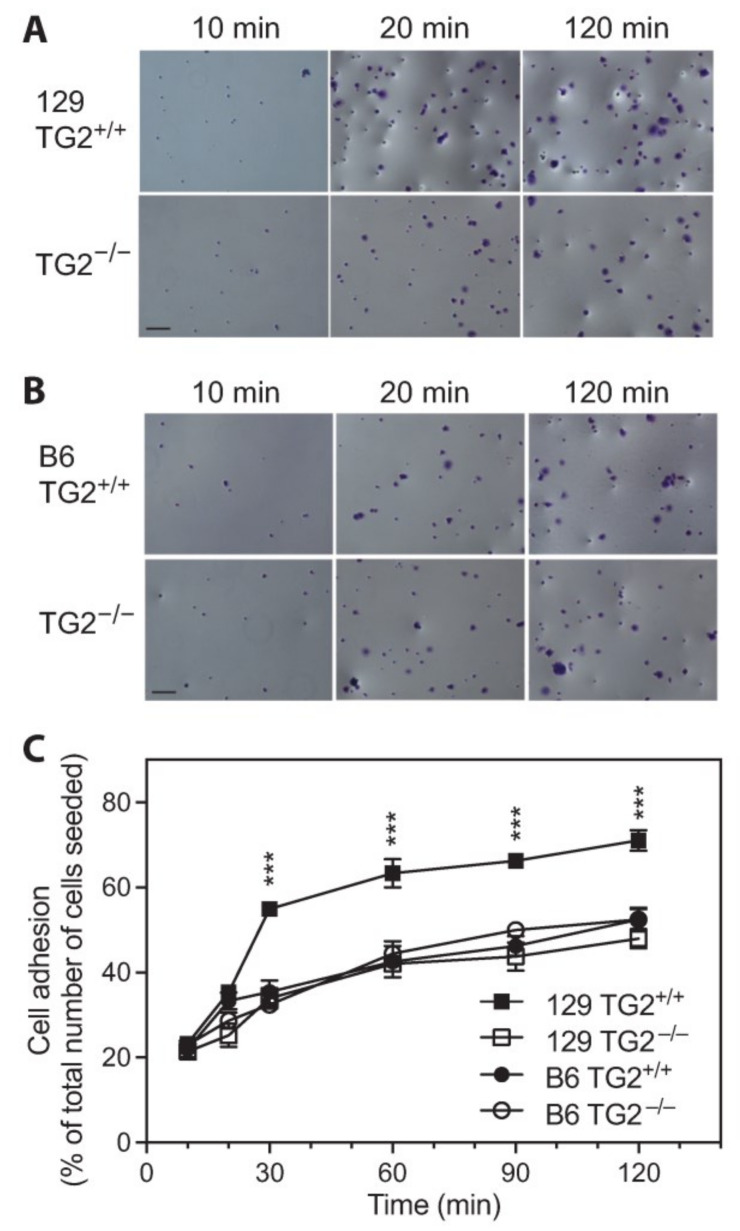
The number of 129 TG2^−/−^ MEFs adherent to Fn-coated plates was less compared to 129 TG2^+/+^ MEFs but was no different from B6 TG2^+/+^ or B6 TG2^−/−^ MEFs. (**A**,**B**) Representative photographs of fixed crystal violet-stained 129 (**A**) or B6 (**B**) TG2^+/+^ or TG2^−/−^ MEFs adhered to plates coated with Fn (10 μg/cm^2^) after 10, 20 and 120 min; 10× magnification, scale bar = 30 μm. (**C**) Quantitation of adherent 129 TG2^+/+^, 129 TG2^−/−^, B6 TG2^+/+^ and B6 TG2^−/−^ MEFs after 10, 20, 30, 60, 90 and 120 min incubation on Fn-coated plates, expressed as a percentage of the total number of cells seeded (*n* = 4 experiments performed in triplicate). ***, *p* < 0.001 for individual post hoc Bonferroni test results from two-way ANOVA comparing 129 TG2^+/+^ with 129 TG2^−/−^, B6 TG2^+/+^ or B6 TG2^−/−^ MEFs.

**Figure 7 ijms-24-11475-f007:**
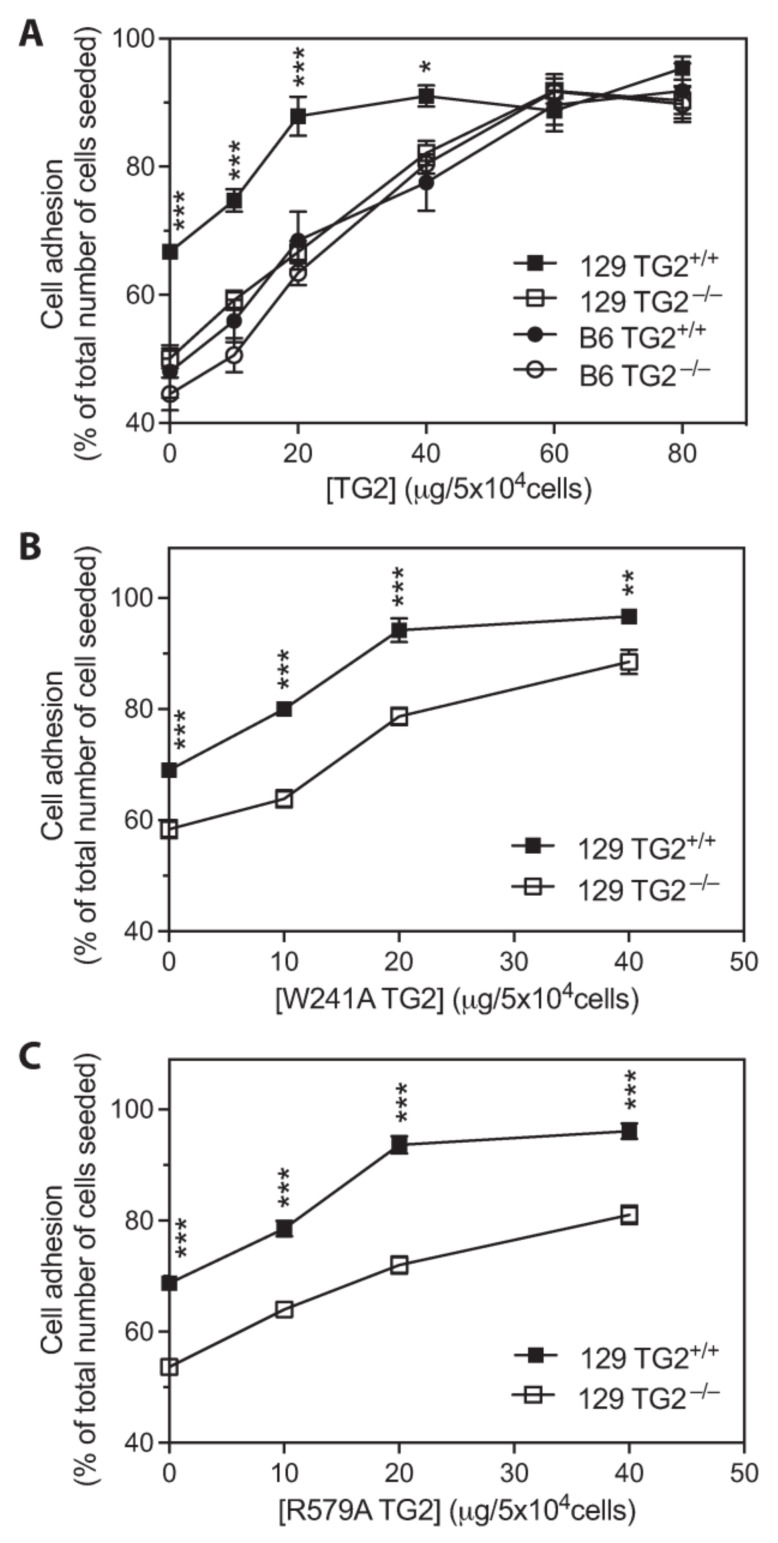
The addition of recombinant wild-type, transamidase-deficient or GTP-binding-deficient TG2 to Fn-coated plates increased the number of adherent MEFs. (**A**–**C**) Quantitation of adherent 129 TG2^+/+^, 129 TG2^−/−^ and (**A**) B6 TG2^+/+^ and B6 TG2^−/−^ MEFs, after 30 min incubation on plates coated with Fn (10 μg/cm^2^) plus (**A**) wild-type (0, 10, 20, 40, 60, 80 μg/cm^2^), (**B**) W241A transamidase-deficient (0, 10, 20, 40 μg/cm^2^) or (**C**) R579A GTP-binding-deficient (0, 10, 20, 40 μg/cm^2^) TG2, expressed as a percentage of the total number of cells (5 × 10^4^) seeded (*n* = 5–6 experiments performed in triplicate). *, *p* < 0.05; **, *p* < 0.01, ***, *p* < 0.001 for individual post hoc Bonferroni test results from two-way ANOVA comparing 129 TG2^+/+^ with 129 TG2^−/−^, B6 TG2^+/+^ or B6 TG2^−/−^ MEFs.

**Figure 8 ijms-24-11475-f008:**
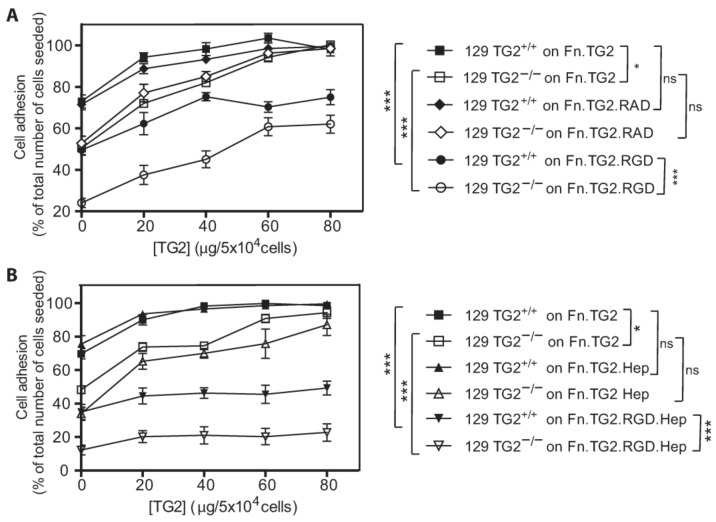
Adhesion of 129 TG2^+/+^ or 129 TG2^−/−^ MEFs to a TG2-Fn matrix was dependent on syndecan binding in the presence, but not absence, of RGD peptides, and the exogenous addition of recombinant wild-type TG2 to an Fn matrix rescued RGD-impaired adhesion of 129 TG2^+/+^ and 129 TG2^−/−^ MEFs. (**A**,**B**) Quantitation of adherent 129 TG2^+/+^ or 129 TG2^−/−^ after 30 min incubation on plates coated with Fn (10 μg/cm^2^) and wild-type TG2 (0, 10, 20, 40, 60, 80 μg/cm^2^) in the absence or presence of (**A**) RGD or RAD peptides (84 μg/cm^2^) or (**B**) heparin (Hep, 42 μg/cm^2^) or heparin and RGD peptide (84 μg/cm^2^), expressed as a percentage of the total number of cells (5 × 10^4^) seeded (*n* = 4 experiments performed in triplicate). ns, *p* > 0.05, *, *p* < 0.05; ***, *p* < 0.001 for two-way ANOVA.

**Figure 9 ijms-24-11475-f009:**
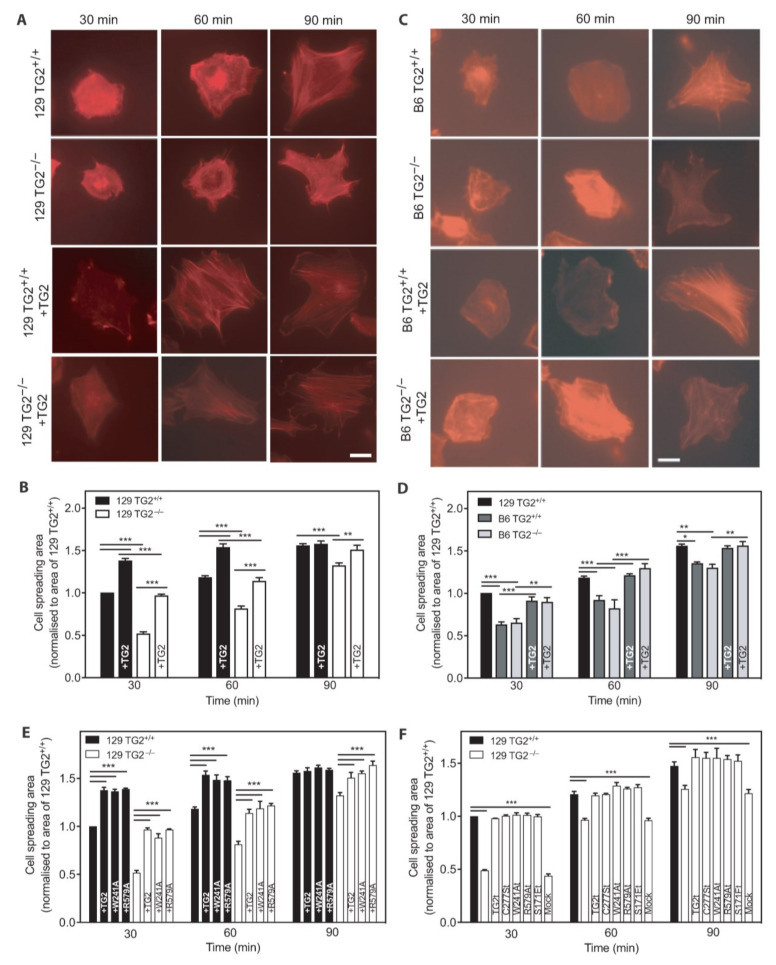
Spreading of 129 TG2^−/−^, B6 TG2^+/+^ and B6 TG2^−/−^ MEFs was equivalent but delayed relative to 129 TG2^+/+^ MEFs and the exogenous addition of recombinant TG2, independent of transamidase or GTP-binding activity, accelerated spreading in all genotypes. (**A**,**C**) Representative photographs of phalloidin-stained actin stress fibers in (**A**) 129 or (**C**) B6 TG2^+/+^ or TG2^−/−^ MEFs after 30, 60 or 90 min incubation on plates coated with Fn (10 μg/cm^2^) without or with recombinant wild-type TG2 (+TG2, 20 μg/cm^2^) added; scale bar = 10 μm. (**B**,**D**,**E**) Quantitation of adherent (**B**,**E**) 129 TG2^+/+^ or 129 TG2^−/−^ and (**D**) B6 TG2^+/+^ or B6 TG2^−/−^ MEF cell areas after 30, 60 or 90 min incubation on plates coated with Fn in the absence or presence of (**B**,**D**,**E**) added recombinant wild-type TG2 (+TG2) or (**E**) added recombinant W241A transamidase-deficient or R579A GTP-binding-deficient TG2, normalized to mean 129 TG2^+/+^ cell area at 30 min post-seeding (*n* = 4–5 experiments performed in triplicate, 10 random cells quantitated per treatment per experiment). (**F**) Quantitation after 30, 60 or 90 min incubation on Fn-coated plates of cell areas of adherent 129 TG2^−/−^ MEFs that were untransfected, or transfected with cDNAs encoding wild-type TG2 (TG2t), transamidase-deficient TG2 (C277St, W241At), GTP-binding-deficient TG2 (R579At, S171Et) or empty vector (Mock), normalized to mean 129 TG2^+/+^ cell area at 30 min post-seeding (*n* = 5 experiments performed in triplicate, 10 random cells quantitated per treatment per experiment). *, *p* < 0.05; **, *p* < 0.01, ***, *p* < 0.001 for individual post hoc Bonferroni test results from two-way ANOVA.

**Figure 10 ijms-24-11475-f010:**
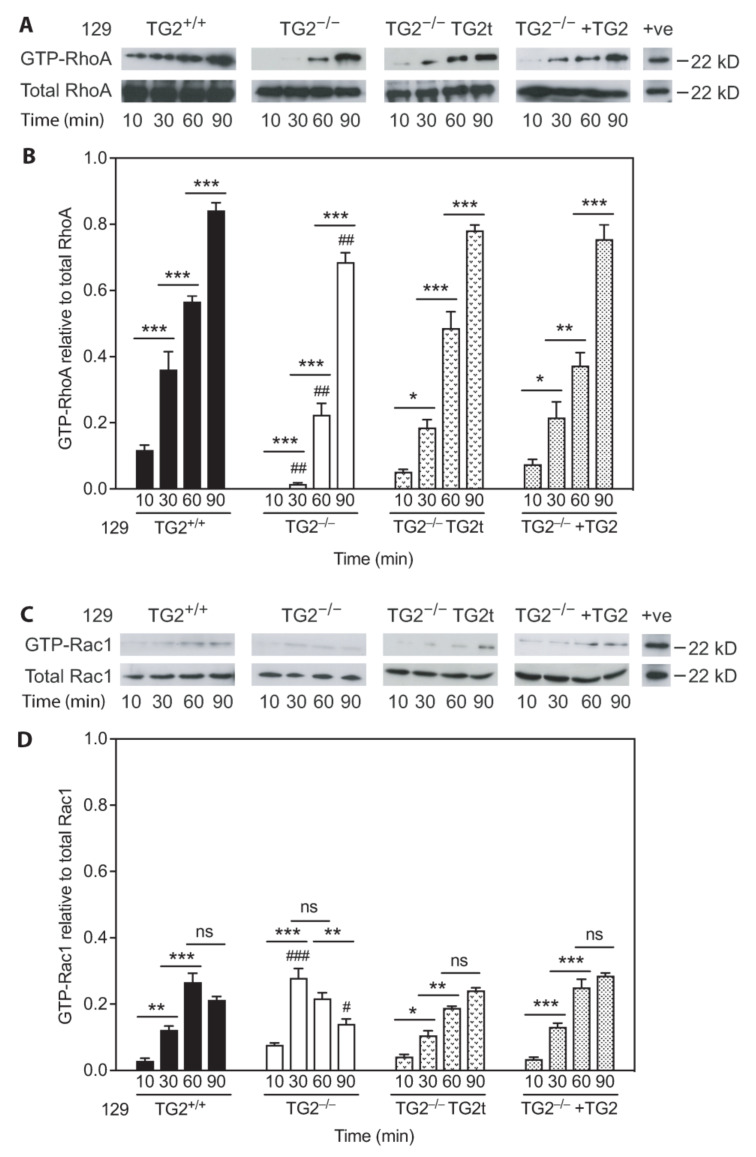
Activation dynamics of the GTPases RhoA and Rac1 that control cell adhesion and spreading on Fn were altered in 129 TG2^−/−^ relative to TG2^+/+^ MEFs. (**A**,**C**) Representative western blots and (**B**,**D**) quantitation of activated GTP-bound RhoA (GTP-RhoA) relative to total RhoA (**B**) or activated GTP-bound Rac1 (GTP-Rac1) relative to total Rac1 (**D**) in lysates from Fn-adherent 129 TG2^+/+^ MEFs, TG2^−/−^ MEFs, TG2^−/−^ MEFs transfected with TG2 cDNA (TG2t) or TG2^−/−^ MEFs with recombinant wild-type TG2 added to the Fn matrix (+TG2, 20 μg/cm^2^) at 10, 30, 60 or 90 min post-seeding, with GTPγS-treated TG2^+/+^ lysates as a positive control (+ve) (*n* = 3 experiments). *, *p* < 0.05; **, *p* < 0.01, ***, *p* < 0.001 for individual post hoc Bonferroni test results from two-way ANOVA. #, *p* < 0.05; ##, *p* < 0.01, ###, *p* < 0.001 for individual post hoc Bonferroni test results two-way ANOVA comparing TG2^−/−^ with TG2^+/+^ MEFs at the indicated time points.

**Figure 11 ijms-24-11475-f011:**
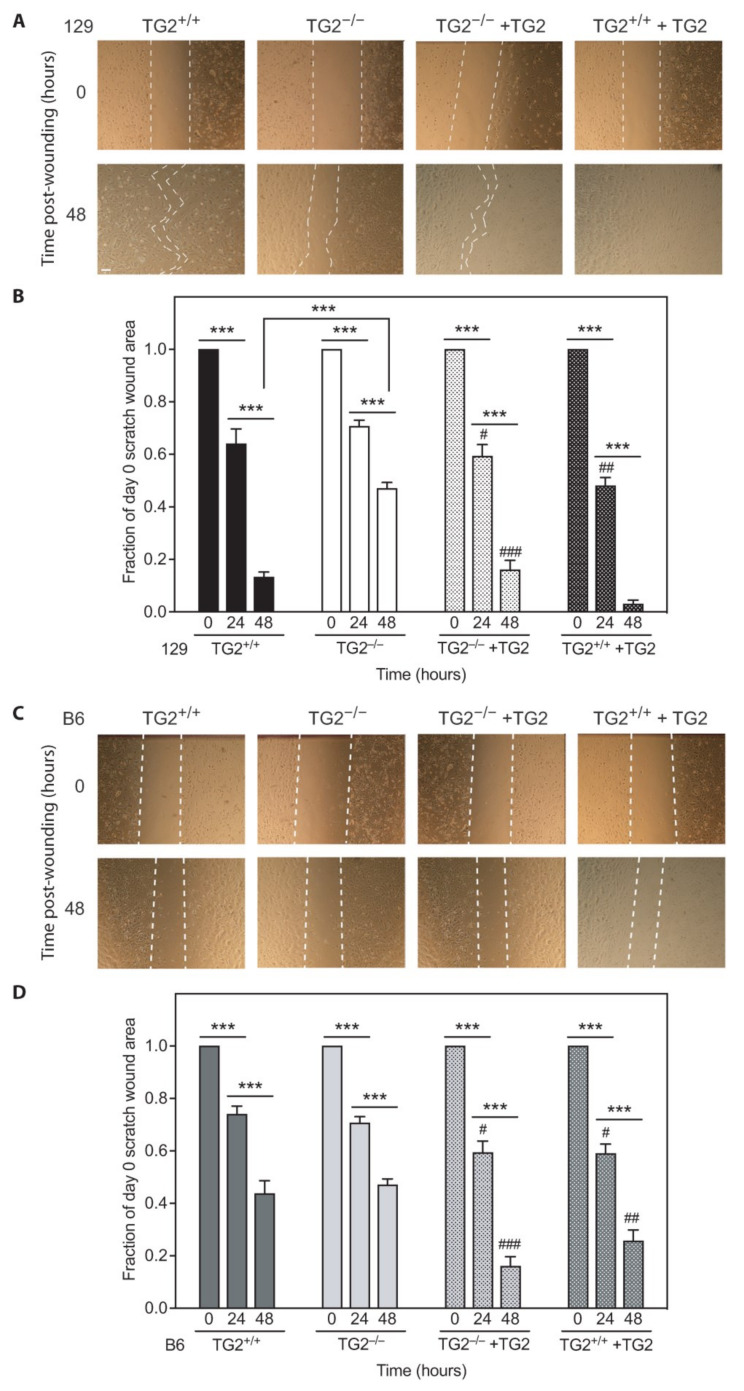
Closure of an in vitro scratch wound in confluent cell monolayers was delayed in 129 but not B6 TG2^−/−^ MEFs, relative to their TG2^+/+^ counterparts, and the exogenous addition of recombinant wild-type TG2 immediately post-wounding accelerated wound closure in all genotypes. Confluent monolayers of (**A**,**B**) 129 TG2^+/+^ or TG2^−/−^ MEFs or (**C**,**D**) B6 TG2^+/+^ or TG2^−/−^ MEFs were scratch wounded and the scratch wound area was quantitated over time in the absence or presence (+TG2) of exogenous addition of recombinant wild-type TG2 immediately post-wounding; representative images (**A**,**C**) and quantitation of scratch wound area as a fraction of day 0 area (**B**,**D**) (*n* = 3 experiments); scale bar = 100 μm. ***, *p* < 0.001 for individual post hoc Bonferroni test results from two-way ANOVA. #, *p* < 0.05; ##, *p* < 0.01, ###, *p* < 0.001 for individual post hoc Bonferroni test results from two-way ANOVA comparing MEFs + TG2 with their respective MEFs at the indicated time points.

**Figure 12 ijms-24-11475-f012:**
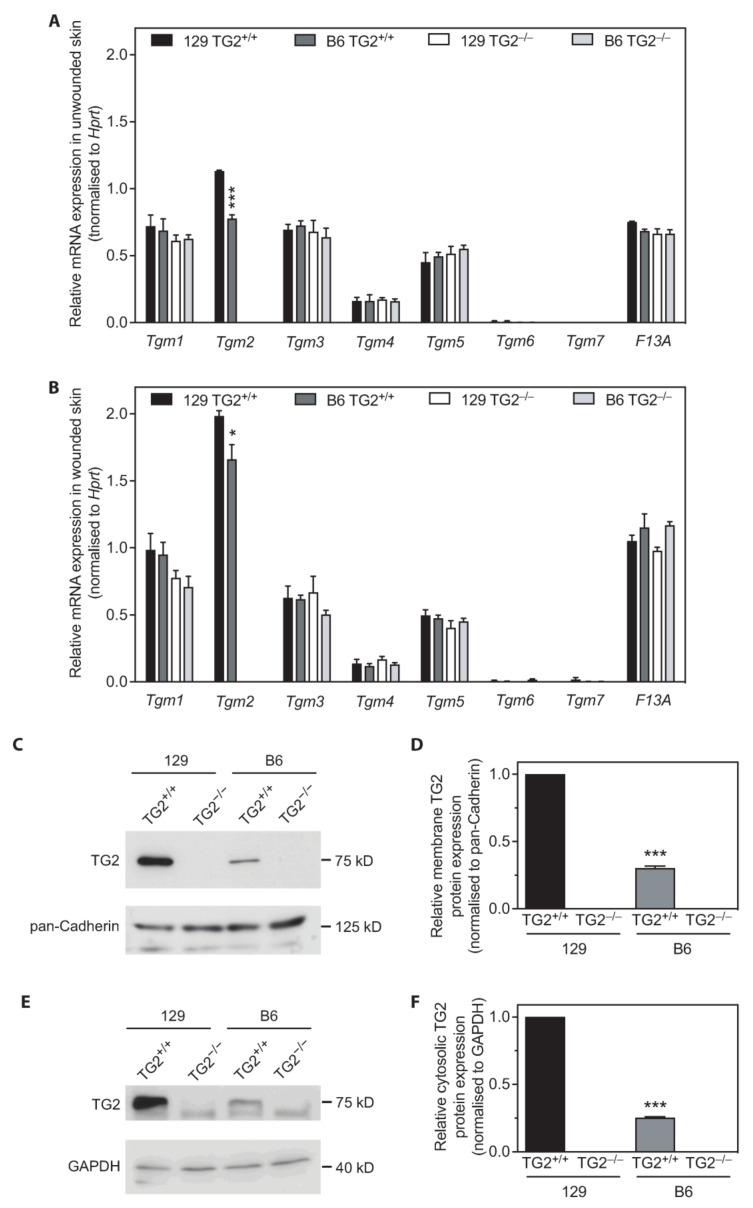
TG2 expression levels are higher in 129 than in B6 mice. (**A**,**B**) cDNA generated from total RNA isolated from 129 or B6 TG2^+/+^ and TG2^−/−^ unwounded skin samples (*n* = 3 per genotype) (**A**) or skin samples on day 2 post-wounding (*n* = 3 per genotype) (**B**) was examined in triplicate for the abundance of mRNAs for TG family members: *Tgm1* (encoding TG1), *Tgm2* (encoding TG2), *Tgm3* (encoding TG3), *Tgm4* (encoding TG4), *Tgm5* (encoding TG5), *Tgm6* (encoding TG6), *Tgm7* (encoding TG7), *f13a* (encoding F13A), normalized to *Hprt* (encoding hypoxanthine phosphoribosyltransferase 1) as the reference gene (expression level unaffected by the experimental treatment). *, *p* < 0.05; ***, *p* < 0.001 for individual post hoc Bonferroni test results from two-way ANOVA comparing 129 TG2^+/+^ with B6 TG2^+/+^ samples. (**C**–**F**) Representative western blots (**C**,**E**) and quantitation (**D**,**F**) of TG2 abundance relative to the reference proteins (expression level unaffected by genotype) pan-Cadherin (**C**,**D**) or GAPDH (**E**,**F**) in membrane or cytosolic fractions, respectively, of 129 or B6 TG2^+/+^ and TG2^−/−^ MEF lysates (*n* = 4 experiments).

## Data Availability

The datasets used and/or analyzed during the current study are available from the corresponding author on reasonable request.

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
