# Peer review of "Transglutaminase 2 Facilitates Murine Wound Healing in a Strain-Dependent Manner"

_ijms, 2023, doi:10.3390/ijms241411475_

Round 1

Reviewer 1 Report

1. The cell area of a single cell is not convincing. It is best to display multiple cells in a large field of view.

2. In WesternBlot experiments, there are differences in the signal strength of individual total proteins, which may affect the accuracy of normalized target protein expression results

3. In in vitro scratch experiments, it is crucial to keep the initial scratch width consistent and vertical, as this ensures consistency of experimental conditions and accuracy of results.

4. When conducting the experiment of wound healing in mice subjected to a circular skin punch biopsy wound, it is essential to ensure consistency in the size and shape of the wounds.

5. Background descriptions for wound healing can be strengthened by citing 10.1016/j.cej.2023.141852; 10.1021/acsmacrolett.2c00290 and what are the advantages of the current work compared to published articles?

6. It is advised to further investigate the signal pathways related to cell cytoskeleton by examining the impact of TG2 gene knockout on the adhesion and extension abilities of other cell types, such as epithelial cells and leukocytes.

7. Explore the potential clinical applications of TG2 by regulating its expression level or applying TG2 agonists or inhibitors and observing their effects on wound healing speed and quality.

ns

Reviewer 2 Report

I really like this study. It provides in depth insight into a specific process involved in wound healing. It is well written, properly executed with extensive supporting experiments. For example, the microarray clearly confirms there is only an effect on Tgm2. Absence of TG2 does only affects function (of fibroblasts) and not overall expression levels of various cells. The fenotype can be restored by addition of TG2 protein.

The most important remark I would like to make, is the distinction between healing and contraction. In the current study, there was an effect on contraction (the movement of skin edges towards the centre of the wound by the action of for example myofibroblasts) and not healing (regeneration of dermal and epidermal tissues). Although this aspect is mentioned (line 95), it should be stated clearly throughout the paper that it is not healing that was studied. But that doesn’t change the relevance of this study, TG2 has a clear effect (in strain 129) on contraction through the ability of fibroblasts to adhere and migrate.

Other issues:

Last part of the introduction (line 72-86) is not introduction but a broad summary of the results.

Figure 1. The quality of the macroscopic wound images must be improved. Why are the wounds not circular at day 0? Error bars should represent standard deviations.

 2.2 In animal experiments (and human wound care) use of analgesics is standard. Omitting analgesics will impact animal welfare and maybe healing. How was this resolved?

 Fig 4. Why was tgfb2 included and not tgfb1? According to this figure hprt1 was the only reference gene. It is however better to use 3 different reference genes. See for example this paper: Reference genes in real-time PCR. J Appl Genet. 2013; 54(4): 391–406. PMID: 24078518 by Kozera and Rapacz. They stated that “experimental confirmation of the stability of candidate genes is now a standard requirement.”

 Since TG2 affects adhesion and migration of fibroblasts, it would be interesting to check tissue samples for fibroblast numbers.

2.6 why were embryonic and not adult fibroblasts used?

Figure 7. what could be the explanation for the increase for 129 +/+ in 7A? These cells should produce sufficient amounts of TG2 themselves?

4.4 please provide source, acclimatization period, individual or grouped housing, analgesics and total number of animals. Were animals daily sedated for the daily wound area measurement? How was the cover slip applied over a curved surface?

4.5 why not use immunohistochemistry for quantification of specific cell types. For example F4/80 & MPO, which are more sensitive and specific. Also, do the reported monocyte numbers in H&E include macrophages and dendritic cells as well?

 4.16 SEM must not be used: the Standard Error of the Mean of the sample is an estimate of how far the sample mean is likely to be from the population mean, whereas the standard deviation (SD) of the sample is the degree to which individuals within the sample differ from the sample mean. A student’s T test can only be used after testing for normality.
